# Emergent Robust Communication for Multi-Round Interactions in Noisy Environments

## Abstract

We contribute a novel multi-agent architecture capable of learning a discrete communication protocol without any prior knowledge of the task to solve. We focus on ensuring agents can create a common language during their training to be able to cooperate and solve the task at hand, which is one of the primary goals of the emergent communication field. On top of this, we focus on increasing the task's difficulty by creating a novel referential game, based on the original Lewis Game, that has two new sources of complexity: adding random noise to the message being transmitted and the capability for multiple interactions between the agents before making a final prediction. When evaluating the proposed architecture on the newly developed game, we observe that the emerging communication protocol's generalization aptitude remains equivalent to architectures employed in much simpler and elementary games. Additionally, our method is the only one suitable to produce robust communication protocols that can handle cases with and without noise while maintaining increased generalization performance levels.

## 1 Introduction

Emergent communication focuses studying the human language's evolution and conceptualizing artificial languages for human-robot communication. Furthermore, with recent advances in deep learning, emergent communication has received special attention from the machine learning community to create learning experiments with neural agents that learn how to communicate without any prior knowledge. From this research standpoint, most works explore emergent communication through a reference-based game called the *Lewis Game* (LG) (Lewis, 1979). In this game, the *Speaker* describes an object to the *Listener*, which has then to discriminate it given a set of candidates. Usually, the communication protocols learned by neural agents when playing the LG appear to be degenerate and lack some core properties of human languages (Rita et al., 2020; Korbak et al., 2020; Chaabouni et al., 2019). Such an outcome is an effect of applying several constraints and simplifications to the Lewis Game to alleviate task complexity to facilitate its modeling as a machine learning problem. For instance, Choi et al. (2018) simplifies the LG by having the Listener describing only two images, and the images have only two effective degrees of freedom. Ren et al. (2020) developed an iterated learning strategy for the LG, intending to create highly compositional languages by creating an algorithm with several prolonged distinct learning phases. Chaabouni et al. (2022) focused on understanding how a population of agents can interfere with the emergence of a communication protocol. This work also scaled some properties of the LG, such as the dataset size used and the number of candidates. However, the authors assume that the Listener always receives information about the correct candidate during the learning phase.

In this paper, we use the work of Chaabouni et al. (2022) as a starting point and lift some of the constraints imposed by the works discussed in the previous paragraph to design more general games, which we can view as a step forward to emulate human communication. In particular, we propose two novel changes to the original LG. First, we add a noisy communication channel where the Speaker's message can suffer unexpected modifications. Second, we also add a time dependency to the LG, where the Listener can decide to play another round of the game or make a final choice.

With these new extensions, we create a more challenging and complete version of the LG to study how a language can emerge in such conditions and also how the properties of the emerged language fundamentally depend on the environment. As a case point, since the broadcasted messages

have some level of uncertainty (introduced by the noise), we can extrapolate this problem as a memory limitation constraint where it becomes impractical for the Listener to memorize specific patterns (Galke et al., 2022). We note that Ueda & Washio (2021) also introduced a setup with noise in the communication channel, using an adversarial setting, to analyze a specific language property: Zipf's law of abbreviation (Zipf, 1999). Conversely, our work focuses on a different axis, which centers on creating robust communication protocols where the pair of agents can communicate with different levels of noise. Furthermore, the authors also implement a simpler version of LG where one-hot vectors are used as input, contrary to real-world images (our case), making the training and architecture design more straightforward.

Additionally, introducing a time dependency to the LG also makes the Listener's decision process more challenging, diminishing the effect of having a large model capable of memorizing specific message patterns instead of language concepts. Ultimately, we expect this pathway to promote compositionality and generalization. As a reference, Bogin et al. (2018) also developed an environment where communication happens during long-horizon tasks. Contrary to our architecture, the authors constrained the Speaker's learning procedure to explicitly output similar messages in related conditions to be able to create functional emerging languages.

We introduce a novel two-agent architecture to solve the games described above. Our method relies on a new approach where both agents, Speaker and Listener, are modeled as reinforcement learning (RL) agents. For our novel Listener architecture, we add a meta-action to model the Listener's decision to play another round instead of making a final decision. With this new action, we allow the Listener to reflect on the current state of the game and the information obtained to decide whether it is best to gather more information or attempt to predict the correct candidate. Our results show that agents trained in the new game learn robust communication protocols to deal with deterministic or noisy messages without losing their general properties. The same does not happen when the agents train in the original LG, where the emerging language cannot handle any noise level.

To summarize, our contributions are 3-fold. First, we introduce a novel game extending the original LG, where the communication channel can suffer interference, implying that some parts of the message can become concealed. Moreover, we extend the LG to have multiple rounds, where more information can flow between the agents before selecting a candidate. We call the introduced game the Multi-Round Indecisive Lewis Game, or MRILG. Second, we develop a new Listener architecture to play the MRILG. When acting, the new architecture deliberates with current and previous round information, choosing whether to play another round or make a final prediction. Third, we evaluate the generated languages in the original and novel LG games regarding their generalization capabilities and transfer learning competence to new tasks. Our results show that the languages are intrinsically different, where simple modifications to the game, like adding stochasticity, can improve the language being learned, allowing for better generalization.

## 2 METHODOLOGY

We start this section by describing the original LG. Next, we explain how to extend the LG to create the proposed and more challenging MRILG. The main changes feature a noisy communication channel and a loop to play the game across multiple rounds, significantly increasing the complexity of the environment. Afterward, we detail how the Speaker converts the received input into a message, a sequence of discrete tokens, and how the Listener processes and integrates the message and candidates to make decisions. We impose a RIAL setting (Foerster et al., 2016), where agents are independent and perceive the other as part of the environment. Hence, we describe the learning strategy for both agents independently, focusing on explaining the loss composition and the importance of each loss term to guide training where functional communication protocols can emerge.

### 2.1 LEWIS GAME (LG)

The Lewis Game (LG) is a discrimination game where one of the agents, the *Speaker*, must describes an object by sending a message to the other agent, the *Listener*. When the game starts, the Speaker receives a target image $x$ retrieved from a fixed dataset $\mathbb{X}$. The Speaker intends to describe the image by encoding a message $m$, a sequence of $T$ discrete tokens, $m(x; \theta) = (u_t(x; \theta))_{t=1}^{T}$. We define $m : \mathbb{X} \to \mathbb{W}^T$, where $\mathcal{W}$ is a finite vocabulary set, and $u_t : \mathbb{X} \to \mathbb{W}$ is a symbol of the vocabulary.

Subsequently, the Listener receives the message along with a set of candidate images, $\mathbb{C} \subset \mathbb{X}$, where the goal is to try to identify the image $\boldsymbol{x} \in \mathbb{C}$ that the Speaker received, $\hat{\boldsymbol{x}} = \boldsymbol{x}$. Both agents receive a reward of $1$ if the Listener is correct, and $-1$ otherwise:

$$R(\boldsymbol{x}, \hat{\boldsymbol{x}}) = \begin{cases} 1, & \text{if } \hat{\boldsymbol{x}} = \boldsymbol{x} \\ -1, & \text{otherwise.} \end{cases}$$

When there is no ambiguity, we drop the dependence of for $\boldsymbol{m}\,(\boldsymbol{x}; \theta)$ on $\boldsymbol{x}$ and $\theta$. Furthermore, the Listener's choice $\hat{\boldsymbol{x}}$ depends on $\boldsymbol{m}$, $\mathbb{C}$, and $\phi$, where $\phi$ contains the Listener's parameters. However, such dependencies will also be omitted for legibility.

## 2.2 MULTI-ROUND INDECISIVE LEWIS GAME (MRILG)

We now introduce and propose a novel extension of the LG, the Multi-Round Indecisive Lewis Game (MRILG). MRILG is composed of several iterative LG rounds. There are at most $N$ rounds to prevent infinite games. At each round, $i \in \{0, \dots, N-1\}$, and similarly to the LG, the Speaker receives a target image $\boldsymbol{x}$ and describes it to the Listener by sending a message composed of discrete tokens, $\boldsymbol{m}\,(\boldsymbol{x}; \theta) = (u_t\,(\boldsymbol{x}; \theta))_{t=1}^{T}$. MRILG has a noisy communication channel, meaning the message can suffer random perturbations when being transmitted. We define a new function that converts each token, with a given probability, into a default unknown token, $\text{noise} : \mathbb{W}^T \to \mathbb{W}'^T$, where $\mathbb{W}'$ contains the original vocabulary $\mathbb{W}$ plus the unknown token, $\mathbb{W}' = \mathbb{W} \cup \{\texttt{unk}\}$. Accordingly, we define a new function to disrupt the message before giving it to the Listener:

$$\text{noise}\,(\boldsymbol{m}) = (n_t\,(u_t\,(\boldsymbol{x}; \theta)))_{t=1}^{T}$$
$$\text{s.t. } n_t\,(u_t\,(\boldsymbol{x}; \theta)) = \begin{cases} u_t(\boldsymbol{x}; \theta), & \text{if } \text{p} > \lambda \\ \texttt{unk}, & \text{otherwise,} \end{cases} \tag{1}$$

where p is sampled from a uniform distribution, $\text{p} \sim \mathcal{U}\,(0, 1)$, and $\lambda$ is a fixed threshold, indicating the noise present in the communication channel. By definition, the Speaker is agnostic to this process and will never know if the message was modified.

Subsequently, the Listener receives the modified message, $\text{noise}\,(\boldsymbol{m})$, and the candidate images, $\mathbb{C}$. Due to the recurrent nature of MRILG, the Listener must be able to leverage different types of actions, thus simulating hierarchical reasoning. Namely, the Listener can decide whether it is preferable to play another round, to gather more information, or to try to predict the correct candidate when anticipates a positive tradeoff between the likelihood of success and the expected reward.

When the Listener makes a final prediction $\boldsymbol{x} \in \mathbb{C}$, both agents receive a positive reward of $1$ if the Listener is correct in its prediction and $-1$ otherwise. Alternatively, the agents proceed to a new round when the Listener plays the *I don't know* (*idk*) action, $\hat{\boldsymbol{x}} = \hat{\boldsymbol{x}}_{\text{idk}}$. We carefully design the reward associated with the *idk* action to avoid degenerate and exploitable cases. For instance, when the *idk* reward tends to the value of the failure reward, the Listener will always attempt to make a final prediction in the first round to avoid huge penalizations. On the other hand, when the *idk* reward exceeds $0$, the Listener gets an incentive to wait for the last round to increase the cumulative reward, even if it knows the correct answer. Additionally, when the gap between the *idk* and failure reward surpasses a certain threshold the Listener will also always play *idk*. In these cases no communication protocol emerges since the Listener completely ignores the Speaker's message.

To tackle all cases above, we define the reward associated with the *idk* action as a negative reward, $\nu \in (-1, 0)$. In Section 3, we explore and compare different values for $\nu$, focusing on how they influence the properties of the emerging language. We define the reward function for the MRILG as:

$$R(\boldsymbol{x}, \hat{\boldsymbol{x}}, i) = \begin{cases} 1, & \text{if } \hat{\boldsymbol{x}} = \boldsymbol{x} \\ \nu, & \text{if } \hat{\boldsymbol{x}} = \hat{\boldsymbol{x}}_{\text{idk}} \text{ and } i < N - 1 \\ -1, & \text{otherwise,} \end{cases}$$

Note that playing the *idk* action in the last round is the same as making a wrong prediction.

### 2.2.1 AGENT ARCHITECTURES

We now describe the architectures implemented for both agents, the Speaker and the Listener (Figure 1). We design the Speaker agent as an RL, where its network architecture is similar to that

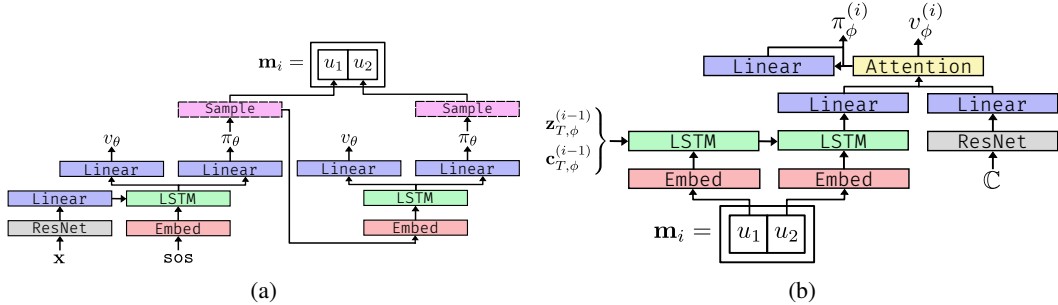

Figure 1: Graphical representation of Speaker, Figure 1a, and Listener, Figure 1b, architectures for the MRILG. In this illustration, the message, $\boldsymbol{m}_i$, contains only two tokens, $T = 2$.

of Chaabouni et al. (2022). In the case of the Listener, we define a novel RL architecture instead of the original Self-Supervised (SS) architecture.

As described in Section 2.1, the Speaker is agnostic to the current game's round $i$. For this reason, we will describe the Speaker's inference and architecture for one game round. As an overview, the Speaker's objective is to encode a discrete message, $\boldsymbol{m}$, describing an input image, $\boldsymbol{x}$. First, we encode the image using a pre-trained image encoder (Grill et al., 2020), $f$, to reduce its dimensionality and extract valuable features, $f(\boldsymbol{x})$. Subsequently, a trainable encoder $g$ processes the new sequence of features outputting the initial hidden and cell values, $(\boldsymbol{z}_{0,\theta}, \boldsymbol{c}_{0,\theta}) = g(f(\boldsymbol{x}); \theta)$, used by the recurrent module $h$, an LSTM. Since $f$ and $g$ are deterministic, $\boldsymbol{z}_{0,\theta}$ and $\boldsymbol{c}_{0,\theta}$ always have the same values when providing the same input image, independently of the game round.

Subsequently, the Speaker will select each token to add to the message iteratively, using $h$. On this account, we define a complementary embedding module, $e$, to convert the previous discrete token $u_{t-1}$ into a dense vector $\boldsymbol{d}_{t,\theta} = e(u_{t-1}; \theta)$. Then, the recurrent module, $h$, consumes the new dense vector and previous internal states to produce the new ones, $(\boldsymbol{z}_{t,\theta}, \boldsymbol{c}_{t,\theta}) = h(\boldsymbol{d}_{t,\theta}, z_{t-1,\theta}, c_{t-1,\theta}; \theta)$. We then pass $\boldsymbol{z}_{t,\theta}$ through two concurrent heads: (i) The actor head yields the probability of choosing each token as the next one, $u_t \sim \pi_S(\cdot|\boldsymbol{z}_{t,\theta})$; (ii) The critic head estimates the expected reward (value function approximation) $V(\boldsymbol{x}) := v(z_{t,\theta}; \theta)$. After the new token is sampled, we feed it back to $e(\cdot; \theta)$, and the process repeats itself until we generate $T$ tokens. The first token $u_0$ is a predefined *start-of-string* token and is not included in the message. Following Chaabouni et al. (2022), we maintain the original vocabulary and message sizes, where $|\mathbb{W}| = 20$, and $T = 10$, which is vastly more extensive than the size of the dataset used $\left(|\mathbb{X}| \approx 10^6\right)$.

The implementation of the Speaker is agnostic to the current game round being played, meaning no information flows between rounds. In contrast, we focus on the listening side by intentionally adding history between rounds and analyzing if the Listener can lead the learning process to more general languages that can effectively play the MRILG.

The Listener architecture has two different modules to process the received message from the Speaker $\boldsymbol{m}$ and the images obtained as candidates $\mathbb{C}$. Additionally, a third module combines the output of both input components and provides it to two heads, actor and critic heads. We now describe each component in detail.

To process the candidate images $\mathbb{C}$, the Listener uses the same pre-trained encoder $f$ combined with the network $c$ to embed the candidates into a lower dimension, $\boldsymbol{l}_{\boldsymbol{x}_j} = c(f(\boldsymbol{x}_j); \phi)$, where $\boldsymbol{x}_j \in \mathbb{C}$.

Concerning the message received each round, $\boldsymbol{m}^{(i)}$, from the communication channel, the Listener uses the recurrent model $h$ (an LSTM) to handle it by processing each token, $u_t^{(i)}$, iteratively. Similarly to the Speaker, there is an embedding layer, $e(\cdot; \phi)$, to convert the discrete token into a dense vector before giving it to the LSTM, where we have $(\boldsymbol{z}_{t,\phi}^{(i)}, \boldsymbol{c}_{t,\phi}^{(i)}) = h(e(u_t^{(i)}; \phi), \boldsymbol{z}_{t-1,\phi}^{(i)}, \boldsymbol{c}_{t-1,\phi}^{(i)}; \phi)$. The initial internal states of $h_\phi$ are initialized as $\boldsymbol{z}_{0,\phi}^{(0)} = \boldsymbol{0}$ and $\boldsymbol{c}_{0,\phi}^{(0)} = \boldsymbol{0}$ for the first round, and for the consecutive rounds are the final hidden and cell values of the previous round, *i.e.* $\boldsymbol{z}_{0,\phi}^{(i)} = \boldsymbol{z}_{T,\phi}^{(i-1)}$ and $\boldsymbol{c}_{0,\phi}^{(i)} = \boldsymbol{c}_{T,\phi}^{(i-1)}$. After processing all message tokens received in the current round, the fi-

nal hidden state, $z_{T,\phi}^{(i)}$, is passed through a final network $g$ to output the message's hidden value $\boldsymbol{l}_m^{(i)} = g(z_{T,\phi}^{(i)}; \phi)$. Finally, the generated hidden values for the message and all candidates flow through to the head module.

The first operation in the head module executes a straightforward attention mechanism to combine the obtained message features with each candidate's counterpart. The output includes a value per candidate which we concatenate into a single vector $\boldsymbol{s}^{(i)} = \begin{bmatrix} \boldsymbol{l}_m^{(i)} \cdot \boldsymbol{l}_{x_1} & \dots & \boldsymbol{l}_m^{(i)} \cdot \boldsymbol{l}_{x_{|\mathbb{C}|}} \end{bmatrix}^T$, called the candidates' score. For the actor head, $\boldsymbol{s}^{(i)}$ passes through a linear layer $t(\cdot \;; \phi)$ to compute the score corresponding to the *idk* action, $s_{\text{idk}}^{(i)}$. Therefore, we define the Listener's policy as $\pi_L^{(i)}(\cdot|\boldsymbol{m}^{(i)}, \mathbb{C}) := \pi_L^{(i)}(\cdot|\boldsymbol{s}^{(i)}, s_{\text{idk}}^{(i)})$, which is a valid approximation since $\boldsymbol{s}^{(i)}$ holds information from the current message and candidates, and $s_{\text{idk}}^{(i)}$ has additional knowledge to compute the *idk* action. At the same time, the critic head $v(\cdot \;; \phi)$ receives the same scores $\boldsymbol{s}^{(i)}$ and uses an MLP to estimate the expected cumulative reward for the current round $i$, as detailed in Section 2.2.2.

### 2.2.2 LEARNING STRATEGY

As described at the start of Section 2.2, the agents can only transmit information via the communication channel, which has only one direction: from the Speaker to the Listener. Additionally, agents learn how to communicate following the RIAL protocol, where agents are independent and treat others as part of the environment. As such, we have a decentralized training scheme where the agents improve their own parameters solely by maximizing the game's reward.

To perform well and consistently when playing the MRILG, the Speaker must learn how to utilize the vocabulary to distinctively encode each image into a message to obtain the highest expected reward possible. We use Reinforce (Williams, 1992), a policy gradient algorithm, to train the Speaker. Given a target image $\boldsymbol{x}$ and game round $i$ with the corresponding Listener's action $\hat{\boldsymbol{x}}^{(i)}$, we have a loss, $L_A^{(i)}$, to fit the actor's head and another one, $L_C^{(i)}$, for the critic's head. We set $L_A^{(i)}(\theta) = -\sum_{t=1}^{T} \text{sg}(R(\boldsymbol{x}, \hat{\boldsymbol{x}}^{(i)}, i) - v(z_{t,\theta}^{(i)}; \theta)) \cdot \log \pi_S(u_t^{(i)}|z_{t,\theta}^{(i)})$ , where $\text{sg}(\cdot)$ is the *stop-gradient* function, in order to optimize the policy. Regarding the critic loss, we set $L_C^{(i)}(\theta) = \sum_{t=1}^{T}(R(\boldsymbol{x}, \hat{\boldsymbol{x}}^{(i)}, i) - v(z_{t,\theta}^{(i)}; \theta))^2$, to approximate the state-value function $V(\boldsymbol{x})$. We also use an additional entropy regularization term, $L_{\mathcal{H}}^{(i)}$, to make sure the language learned by the Speaker will not entirely stagnate by encouraging new combinations of tokens that increase entropy, further incentivizing exploration. Moreover, we define a target policy for the Speaker to minimize an additional KL divergent term, $L_{\text{KL}}^{(i)}$, between the online and target policies, $\theta$ and $\bar{\theta}$, respectively. We update $\bar{\theta}$ using an exponential moving average (EMA) over $\theta$, *i.e.* $\bar{\theta} \leftarrow (1-\eta)\theta + \eta\bar{\theta}$ where $\eta$ is the EMA weight parameter. With $L_{\text{KL}}^{(i)}$, we prevent steep changes in the parameter space, which helps stabilize training (Rawlik et al., 2012; Chane-Sane et al., 2021). We refer to Chaabouni et al. (2022) for a complete analysis on the impact of $L_{\text{KL}}^{(i)}$. Finally, we weigh each loss term and average the resulting sum for each input image in a batch, $\mathbb{X}' \subset \mathbb{X}$, and *effective* game round, $i \in \{1, \dots, I\}$, to obtain the overall Speaker loss: $L(\theta) = \frac{1}{|\mathbb{X}'|} \sum_{\hat{x} \in \mathbb{X}'} \frac{1}{I} \sum_{i=1}^{I} \alpha_{S,A} L_A^{(i)}(\theta) + \alpha_{S,C} L_C^{(i)}(\theta) + \alpha_{S,\mathcal{H}} L_{\mathcal{H}}^{(i)}(\theta) + \alpha_{S,\text{KL}} L_{\text{KL}}^{(i)}(\theta)$, where $\alpha_{S,A}$, $\alpha_{S,C}$, $\alpha_{S,\mathcal{H}}$, $\alpha_{S,\text{KL}}$ are constants, and the total effective game round, $I \in \{0, \dots, N-1\}$, is the number of rounds played by the pair of agents up to (and including) the first round the Listener plays a decisive action, meaning it tries to predict the correct candidate, $\hat{\boldsymbol{x}}^{(I)} \neq \hat{\boldsymbol{x}}_{\text{idk}}$.

We also use Reinforce to train the Listener agent. Since the Listener preserves state between rounds, we compute the expected cumulative reward as follows:

$$G(\boldsymbol{x}, \hat{\boldsymbol{x}}, i, I) = \sum_{j=i}^{I-1} \gamma^{j-i} R(\boldsymbol{x}, \hat{\boldsymbol{x}}_{\text{idk}}, j) + \gamma^{I-i} R(\boldsymbol{x}, \hat{\boldsymbol{x}}, I). \tag{2}$$

We define $L_A^{(i)}(\phi) = -\text{sg}(G_i(\boldsymbol{x}, \hat{\boldsymbol{x}}, i, I) - v(\boldsymbol{s}^{(i)}; \phi)) \cdot \log(\pi_L^{(i)}(\hat{\boldsymbol{x}}'|\boldsymbol{s}^{(i)}, s_{\text{idk}}^{(i)})$, where $\hat{\boldsymbol{x}}'$ is $\hat{\boldsymbol{x}}_{\text{idk}}$ if $i < I$, and $\hat{\boldsymbol{x}} \in \mathbb{C}$ otherwise. Additionally, we set $L_C^{(i)}(\phi) = (G_i(\boldsymbol{x}, \hat{\boldsymbol{x}}, i, I) - v(s^{(i)}; \phi))^2$, to train the critic head. Similarly to the Speaker loss, we add an entropy loss term $L_{\mathcal{H}}^{(i)}(\phi)$ to

encourage exploration. The final Listener loss for a batch of images and each game round is $L(\phi) = \frac{1}{|\mathbb{X}'|} \sum_{\hat{x} \in \mathbb{X}'} \frac{1}{I} \sum_{i=1}^{I} \alpha_{L,A} L_A^{(i)}(\phi) + \alpha_{L,C} L_C^{(i)}(\phi) + \alpha_{L,\mathcal{H}} L_{\mathcal{H}}^{(i)}(\phi)$, where $\alpha_{L,A}$, $\alpha_{L,C}$, and $\alpha_{L,\mathcal{H}}$ are constants. A detailed analysis of the learning strategy, for both agents, can be found in Appendix E.1.

Due to the complexity and non-stationarity of the environment, we found several requirements to be necessary to guide the training of both agents towards regions in the parameter space where viable communication protocols emerge, instead of being degenerate. Namely, we add a pre-train phase to MRILG, where we linearly increase the noise level in the communication channel from 0 to $\lambda$. This phase is optional and only helps with data efficiency (we refer to Appendix E.3 for more details). We also observe that the Listener benefits from having both actor and critic heads, where the interplay between the actor and critic loss terms is essential to effectively guide both agents' learning in an early training phase. For instance, having these two loss terms reduces the Listener policy's entropy substantially, meaning the Listener learns that there is only one correct action (one candidate) for each round and game. On the other hand, we show that when the Listener architecture comprises only an actor, the agents never agree on a communication protocol, where the training fails entirely. In Appendix E.4, we present an ablation study that corroborates these findings.

## 3 EVALUATION

We provide an extensive evaluation of MRILG and variants. For completeness, we also consider, as a baseline, the original architecture proposed by Chaabouni et al. (2022) to play the original LG. Our model surpasses this baseline at a slight cost of data efficiency. This trade-off is expected and fully explained in Section 3.2.1. At a glance, this happens because the baseline version can access more information than our implementation, during training. We also present and evaluate several intermediary LG variants where we start at the original LG and incrementally modify it until we arrive at the MRILG. Having a progressive sequence of LG games enables us to assess how each environment modification influences the emergent communication protocol learned by the agents.

We continue this section by introducing all LG variants, giving a broader view of each game, agent architectures, and learning strategy. Next, we evaluate the generality of the emerging language for each game variant when providing new and unseen images.

Due to space constraints, we provide additional results in the appendix, where we evaluate all game variants in several transfer learning tasks, named *ETL*. This supplementary evaluation gives yet another frame of reference to evaluate the generality and robustness of the learned languages. Finally, we also refer to the appendix for further implementation details regarding every game variant (Appendix B), model architectures (Appendix D), and datasets used (Appendix E.2).

### 3.1 LEWIS GAME VARIANTS

We briefly report essential aspects of each game variant, and Appendix B presents supplementary information. We consider four variants of the LG, all of which share the same Speaker architecture. The Listener architecture differs in all games. We refer to Appendix C for a comprehensive description of these architectures. Additionally, all variants except for LG (SS) are a contribution of this work. We now describe the LG variants considered:

- LG (SS): The original LG variant of Chaabouni et al. (2022). Here, the Listener is a Self-Supervised agent that tries to find similarities between the received message and the correct candidate through the InfoNCE loss (Oord et al., 2018; Dessi et al., 2021).

- LG (RL): We consider an RL Listener, trained using Reinforce (Williams, 1992). The architecture is similar to that in Section 2.2.1, without computing the *idk* action.

- NLG: In the NLG (which stands for Noisy Lewis Game), we apply an external environmental pressure by adding noise to the message during transmission. The agents' implementation and learning strategy is the same as in the LG (RL).

- MRILG: The most general and the target game introduced in Section 2.2. Note that this is the only game where the *idk* action has an extended impact on the game itself since it allows the agents to play another round.

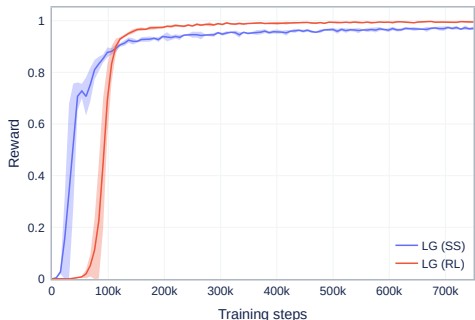

Figure 2: Accuracy during training of LG (SS) and LG (RL) with $|\mathbb{C}| = 1024$ on ImageNet dataset.

## 3.2 ROBUST COMMUNICATION PROTOCOLS

This section thoroughly analyzes the performance of the different variants described above. Starting by comparing LG (SS) with LG (RL), we can see an apparent performance boost for the LG when implementing the Listener as an RL agent. Figure 2 shows that the RL version performs better than the SS version. Equivalent results occur in the testing phase (Table 1), where the RL version surpasses the accuracy achieved by the SS counterpart. From Figure 2, we also observe a trade-off between performance and sample efficiency, where the RL version is less sample efficient. We can trace these differences back to the loss function employed by each version. For instance, the SS version employs the InfoNCE loss, which we can see as a Reinforce variant with only a policy to optimize and, particularly, with access to an oracle, which gives information about which action (candidate) is the right one for each received message. As such, the Listener (SS) can efficiently learn how to map messages to the correct candidates. On the other hand, the RL version has no access to such an oracle and needs to interact with the environment to build this knowledge. The decrease in sample efficiency from SS to RL is, therefore, a natural phenomenon. Nonetheless, the RL version introduces a critic loss term whose synergy with the policy loss term helps to improve the final performance compared to the SS version.

One disadvantage of employing, at inference time, the communication protocols learned by playing default LG variants (LG (SS) and LG (RL)) is that they are not robust to deal with message perturbations. Since agents train only with perfect communication, they never experience noisy communication. The performances of SS and RL with noisy communication thus experience an extensive drop as shown in Table 2.

When introducing noise at training time (NLG), we place the agents in a more complex environment where only random fractions of the message are visible. Despite such modifications, they can still adapt to the environment and learn robust communication protocols that handle both types of messages (with and without noise). We notice equivalent accuracy performance for NLG and LG (RL) when testing with perfect communication channels and a significant boost when conducting tests with noisy communication channels (see Tables 1 and 2, respectively). These results indicate that adding noise at training time is an effective solution to learn more robust communication protocols.

### 3.2.1 COMPARING DIFFERENT LEVELS OF NOISE

Examining the results obtained from NLG and MRILG against LG (RL) when evaluating using a noiseless channel, we observe no loss in inference capabilities for any noise level applied during training, as shown in Table 3.

Additionally, since we force the evaluation game to have at most one round, we observe no clear advantage when training agents in the multi-round game against NLG. From Table 4, we observe that the test accuracy decreases as the noise increases. Additionally, when training with $\lambda = 0.25$, the performance loss in test accuracy is almost negligible compared to the results in Table 3. The agents can effectively sustain this noise threshold without losing performance. One possible reason for this behavior is that messages are overly lengthy, where even having just $75\%$ of the tokens is still enough to encode useful information regarding the Speaker's input.

Table 1: Test accuracy with SD for different game variants, using ImageNet dataset and over 10 seeds. During training $|\mathbb{C}| = 1024$. During test $\lambda_{\text{test}}$ is set to 0.

| Game | $\lambda$ | $|\mathbb{C}|$ (test) | |
|---|---|---|---|
| | | 1024 | 4096 |
| LG (SS) | 0 | 0.96 (0.02) | 0.88 (0.04) |
| LG (RL) | 0 | 0.99 (0.00) | 0.97 (0.00) |
| NLG | 0.5 | 0.99 (0.00) | 0.97 (0.00) |

Table 2: Test accuracy with SD for different game variants, using ImageNet dataset and over 10 seeds. During training $|\mathbb{C}| = 1024$. During test $\lambda_{\text{test}}$ is set to 0.5.

| Game | $\lambda$ | $|\mathbb{C}|$ (test) | |
|---|---|---|---|
| | | 1024 | 4096 |
| LG (SS) | 0 | 0.05 (0.01) | 0.03 (0.01) |
| LG (RL) | 0 | 0.11 (0.02) | 0.06 (0.01) |
| NLG | 0.5 | 0.82 (0.01) | 0.68 (0.02) |

Table 3: Test accuracy with SD for different game variants, using ImageNet dataset and over 10 seeds. During test $\lambda_{\text{test}}$ is set to 0.

| Game | $\lambda$ | $\nu$ | $|\mathbb{C}|$ (test) | |
|---|---|---|---|---|
| | | | 1024 | 4096 |
| NLG | 0.25 | - | 0.99 (0.00) | 0.97 (0.00) |
| NLG | 0.5 | - | 0.99 (0.00) | 0.97 (0.00) |
| NLG | 0.75 | - | 0.98 (0.00) | 0.94 (0.01) |
| MRILG | 0.25 | -0.2 | 0.99 (0.00) | 0.97 (0.01) |
| MRILG | 0.5 | -0.2 | 0.99 (0.00) | 0.97 (0.01) |
| MRILG | 0.75 | -0.2 | 0.98 (0.01) | 0.94 (0.04) |

Table 4: Test accuracy with SD for different game variants, using ImageNet dataset and over 10 seeds. During test $\lambda_{\text{test}}$ is set to $\lambda$.

| Game | $\lambda$ | $\nu$ | $|\mathbb{C}|$ (test) | |
|---|---|---|---|---|
| | | | 1024 | 4096 |
| NLG | 0.25 | - | 0.97 (0.00) | 0.90 (0.01) |
| NLG | 0.5 | - | 0.82 (0.01) | 0.68 (0.02) |
| NLG | 0.75 | - | 0.36 (0.01) | 0.23 (0.01) |
| MRILG | 0.25 | -0.2 | 0.96 (0.01) | 0.89 (0.02) |
| MRILG | 0.5 | -0.2 | 0.80 (0.01) | 0.65 (0.02) |
| MRILG | 0.75 | -0.2 | 0.37 (0.01) | 0.23 (0.01) |

### 3.2.2 COMPARING DIFFERENT REWARDS FOR THE IDK ACTION

Focusing on the *idk* reward's impact, $\nu$, we observe a clear advantage of using a lower value for the *idk* reward, see Tables 5 and 6. We observe similar results when the $\nu = \{-0.5, -0.2\}$. On the other hand, having $\nu$ close to 0 decreases test accuracy considerably. In addition, the training is unstable for this *idk* reward level since a considerably high variance appears in all results. As such, we discern that adding more cost to the *idk* action is a clear strategy to ensure the Listener does not exploit this action and only uses it when there is high uncertainty about the correct candidate.

### 3.2.3 SCALING THE NUMBER OF CANDIDATES

For every different game, we scale the number of candidates, $|\mathbb{C}|$, from 16 to 1024, using a ratio of 4. Looking at Tables 7 and 8, we can see an evident generalization boost when the number of candidates increases for every game. We posit that increasing the game's difficulty (increasing the number of candidates) helps the agents to generalize. As the candidates' set gets additional images, the input diversity increases, which affects how agents encode and interpret more information to distinguish the correct image from all others.

We note that as $|\mathbb{C}|$ increases, the test performance also increases, but at a smaller scale, e.g., the test gap (when at test time $|\mathbb{C}| = 4096$ candidates), between LG (RL) with $|\mathbb{C}| = 16$ and $|\mathbb{C}| = 64$ is 0.4 and only 0.03 between $|\mathbb{C}| = 256$ and $|\mathbb{C}| = 1024$. The noisy games exhibit the same behavior and have comparable results to LG (RL) despite the increased complexity in the environments.

## 4 CONCLUSION & FUTURE WORK

In this work, we focus on creating emergent and robust languages that can be safely employed at inference time since they are robust to perturbation in the communication channel. Following this

Table 5: Test accuracy with SD for different game variants, using ImageNet dataset and over 10 seeds. During test $\lambda_{\text{test}}$ is set to 0.

| Game | $\lambda$ | $\nu$ | $|\mathbb{C}|$ (test) | |
|------|-----------|-------|------|------|
| | | | 1024 | 4096 |
| MRILG | 0.5 | -0.5 | 0.99 (0.00) | 0.96 (0.00) |
| MRILG | 0.5 | -0.2 | 0.99 (0.00) | 0.97 (0.00) |
| MRILG | 0.5 | -0.05 | 0.79 (0.42) | 0.72 (0.38) |

Table 6: Test accuracy with SD for different game variants, using ImageNet dataset and over 10 seeds. During test $\lambda_{\text{test}}$ is set to 0.5.

| Game | $\lambda$ | $\nu$ | $|\mathbb{C}|$ (test) | |
|------|-----------|-------|------|------|
| | | | 1024 | 4096 |
| MRILG | 0.5 | -0.5 | 0.81 (0.01) | 0.66 (0.02) |
| MRILG | 0.5 | -0.2 | 0.80 (0.01) | 0.65 (0.02) |
| MRILG | 0.5 | -0.05 | 0.62 (0.33) | 0.50 (0.26) |

Table 7: Test accuracy with SD for different game variants, using ImageNet dataset and over 10 seeds. During test $\lambda_{\text{test}}$ is set to 0.

| Game | $\lambda$ | $\nu$ | $|\mathbb{C}|$ | $|\mathbb{C}|$ (test) | |
|------|-----------|-------|------|------|------|
| | | | | 1024 | 4096 |
| LG (RL) | 0 | - | 16 | 0.67 (0.04) | 0.39 (0.04) |
| LG (RL) | 0 | - | 64 | 0.93 (0.01) | 0.79 (0.03) |
| LG (RL) | 0 | - | 256 | 0.98 (0.00) | 0.94 (0.01) |
| LG (RL) | 0 | - | 1024 | 0.99 (0.00) | 0.97 (0.00) |
| NLG | 0.5 | - | 16 | 0.55 (0.03) | 0.27 (0.02) |
| NLG | 0.5 | - | 64 | 0.87 (0.01) | 0.67 (0.03) |
| NLG | 0.5 | - | 256 | 0.98 (0.00) | 0.91 (0.01) |
| NLG | 0.5 | - | 1024 | 0.99 (0.00) | 0.97 (0.00) |
| MRILG | 0.5 | -0.5 | 16 | 0.56 (0.04) | 0.29 (0.04) |
| MRILG | 0.5 | -0.5 | 64 | 0.90 (0.01) | 0.72 (0.03) |
| MRILG | 0.5 | -0.5 | 256 | 0.98 (0.00) | 0.92 (0.01) |
| MRILG | 0.5 | -0.5 | 1024 | 0.99 (0.00) | 0.96 (0.00) |

Table 8: Test accuracy with SD for different game variants, using ImageNet dataset and over 10 seeds. During test $\lambda_{\text{test}}$ is set to 0.5.

| Game | $\lambda$ | $\nu$ | $|\mathbb{C}|$ | $|\mathbb{C}|$ (test) | |
|------|-----------|-------|------|------|------|
| | | | | 1024 | 4096 |
| LG (RL) | 0 | - | 16 | 0.03 (0.01) | 0.01 (0.01) |
| LG (RL) | 0 | - | 64 | 0.09 (0.07) | 0.05 (0.03) |
| LG (RL) | 0 | - | 256 | 0.11 (0.06) | 0.06 (0.04) |
| LG (RL) | 0 | - | 1024 | 0.11 (0.02) | 0.06 (0.01) |
| NLG | 0.5 | - | 16 | 0.32 (0.02) | 0.14 (0.01) |
| NLG | 0.5 | - | 64 | 0.57 (0.02) | 0.33 (0.02) |
| NLG | 0.5 | - | 256 | 0.73 (0.01) | 0.54 (0.02) |
| NLG | 0.5 | - | 1024 | 0.82 (0.01) | 0.68 (0.02) |
| MRILG | 0.5 | -0.5 | 16 | 0.33 (0.03) | 0.14 (0.02) |
| MRILG | 0.5 | -0.5 | 64 | 0.59 (0.02) | 0.35 (0.02) |
| MRILG | 0.5 | -0.5 | 256 | 0.75 (0.02) | 0.55 (0.03) |
| MRILG | 0.5 | -0.5 | 1024 | 0.81 (0.01) | 0.66 (0.02) |

motivation, we extend the LG arriving at more complex variations to add robustness to the communication protocol. We add a noisy communication channel during training and enable a more elaborate interaction between agents by allowing the Listener to ask to play another round before making a final decision. These environment modifications and the introduction of a novel Listener architecture permit the emergence of robust languages to noise. These robust communication protocols perform similarly to the original LG using deterministic channels. Additionally, when using noisy channels, the robust communication protocols surpass the performance of the original LG counterparts.

As future work, we could evaluate different types of noise induced in the communication channel, like changing tokens for others of the same vocabulary, or even using an additional agent to control what tokens to hide. One could also employ the agents' architectures in other, more complex, environments with long-horizon rewards, where the Listener intentionally asks for another message from the speaker when evaluating which action to take. Additionally, as we emphasize in the last paragraph of Section 2.2.2, a deeper study is encouraged to understand the conditions necessary for the agents learn how to communicate and agree on a communication protocol.

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
