# OpenReview forum: "Emergent Robust Communication for Multi-Round Interactions in Noisy Environments"
_ICLR.cc/2024/Conference — Submitted to ICLR 2024_

### Official Review · Reviewer_VCLK · 2023-10-30

**Soundness:** 2 fair
**Presentation:** 3 good
**Contribution:** 2 fair
**Rating:** 6
**Confidence:** 3

**Summary:**

This paper proposes a novel environment for emergent communication. It modifies the original Lewis Game by adding random noise to the communication channel and allowing the receiver to wait for multiple rounds before making the decision. In the experiments, communication protocols that emerged in different variants of the Lewis Game are compared. Protocols produced through this new environment are shown to be more robust and generalizable.

**Strengths:**

1. This paper brings attention to a new research direction – robust communication in the emergent communication field. It is an important topic worth exploring. Environments and training frameworks are carefully designed for this new task.

2. A comprehensive suite of evaluations is conducted to cover various aspects of the environment and framework design. Also, different tasks and evaluation metrics are considered.

**Weaknesses:**

1. Multi-round Lewis Game

    a. There are several works introducing the multi-round Lewis Game and its corresponding listener architectures [1,2]. Justification would be better to underscore your contribution by comparing and contrasting these literatures.

    b. From the experimental results (Table 3,4,7,8), the accuracies of game variants with/without multiple rounds do not differ a lot. More elaboration or experiments are expected to ablate the contribution of the multi-round setting.

    [1] Evtimova, Katrina, et al. "Emergent communication in a multi-modal, multi-step referential game." arXiv preprint arXiv:1705.10369 (2017).

    [2] Qiu, Shuwen, et al. "Emergent graphical conventions in a visual communication game." Advances in Neural Information Processing Systems 35 (2022): 13119-13131.

2. Training and testing in the same noisy environment may not be enough to show that “robust communication” emerges:

    a. We consider the noise may come from the observation beside the communication channel, for example from the sender or receiver sides.  [3]

    b. Instead of also replacing message tokens during testing, other interfering methods to the communication channel can be applied, for example randomly dropping tokens in the messages.

    c. What will be the result when the agents are trained with $\lambda=0.75$ and test with 0.5 and vice versa?

    d. How would compare with previous works on adding noise to the communication channel? [4,5]

    [3] Ueda, Ryo, and Koki Washio. "On the relationship between zipf’s law of abbreviation and interfering noise in emergent languages." Proceedings of the 59th Annual Meeting of the Association for Computational Linguistics and the 11th International Joint Conference on Natural Language Processing: Student Research Workshop. 2021.

    [4] Tucker, Mycal, et al. "Emergent discrete communication in semantic spaces." Advances in Neural Information Processing Systems 34 (2021): 10574-10586.

    [5] Kuciński, Łukasz, et al. "Catalytic role of noise and necessity of inductive biases in the emergence of compositional communication." Advances in Neural Information Processing Systems 34 (2021): 23075-23088.

**Questions:**

1. What is the maximum number of round $I$ set to? Throughout the training process, does the average number of communicative rounds vary a lot? For example, from more rounds to fewer rounds? This may help validate your multi-round design.

2. There are several references missing on page 6.

---

> ### Author Response · Authors · 2023-11-23
> **Response to Reviewer VCLK**
>
> ## Weaknesses
>
> > **There are several works introducing the multi-round Lewis Game and its corresponding listener architectures [1,2]. Justification would be better to underscore your contribution by comparing and contrasting these literatures.**
>
> We thank Reviewer VCLK for the suggestions. We added these works to the extended related work section in Appendix A. Nevertheless, there are significant differences between [1,2] and our work: In [1], the Listener processes one candidate at a time, and the vocabulary used contains only two tokens. In our case, our vocabulary has 20 tokens, and the Listener tries to discriminate between all candidates at each round, which substantially increases game complexity. In [2], the game design is similar to ours, where the Listener receives a message and decides to play another round or select a candidate as the final answer. Nonetheless, our approach is far more complex and realistic than in [2] since we have a discrete communication channel that can suffer modifications from an external source instead of a continuous communication channel [2]. Complementarily, [2] simplifies the problem by using a DIAL approach, allowing gradients to flow between agents. In contrast, our method uses a RIAL approach, where agents perceive others as part of the environment, meaning no gradient flows between agents. In this case, the Speaker does not know that the message is being modified, complicating the coordination task.
>
> ---
>
> > **From the experimental results (Table 3,4,7,8), the accuracies of game variants with/without multiple rounds do not differ a lot. More elaboration or experiments are expected to ablate the contribution of the multi-round setting.**
>
> With these results, we observe no clear advantage to using multiple rounds; the agents have enough predictive power to play the game successfully with only one round.
>
> ---
>
> > **Training and testing in the same noisy environment may not be enough to show that “robust communication” emerges: We consider the noise may come from the observation beside the communication channel, for example from the sender or receiver sides. [3]**
>
> We thank Reviewer VCLK for the suggestion. This is an interesting experiment to explore. We already have further experiments in Appendix F; one of them (discrimination task) does this, adding noise to the observation of the Speaker and the Listener's candidates.
>
> ---
>
> > **Instead of also replacing message tokens during testing, other interfering methods to the communication channel can be applied, for example randomly dropping tokens in the messages.**
>
> This is what we do, the noise we introduce acts by replacing message token by a pre-defined token called the *unk* token.
>
> ---
>
> > **What will be the result when the agents are trained with $\lambda=0.75$ and test with 0.5 and vice versa?**
>
> We thank Reviewer VCLK for the suggestion. We can easily add further results considering this (experiments are already running).
>
> ---
>
> > **How would compare with previous works on adding noise to the communication channel? [4,5]**
>
> We thank Reviewer VCLK for the suggestions. We added these works to the extended related work section in Appendix A. Still, there are some major differences. [4] uses a DIAL setting, and the noise is introduced in the continuous space (before sampling discrete tokens). [5] also implements a DIAL setting and the introduction of noise is adversarial, meaning message tokens are changed by other valid tokens, trying to deceive the Listener. In contrast, our method uses a RIAL approach, where agents perceive others as part of the environment, meaning no gradient flows between agents. In this case, the Speaker does not know that the message is being modified, complicating the coordination task.
>
> ## Questions
>
> > **What is the maximum number of round $I$ set to? Throughout the training process, does the average number of communicative rounds vary a lot? For example, from more rounds to fewer rounds? This may help validate your multi-round design.**
>
> Yes, the avg. number of rounds increases as the training progresses. We can add a section to the appendix displaying this information. Although the number of rounds increases, the overall performance remains the same.
>
> ---
>
> > **There are several references missing on page 6.**
>
> This was a problem when splitting the document into 2 parts (main paper and appendix). We fixed the problem, thank you.

---

### Official Review · Reviewer_zpnz · 2023-10-30

**Soundness:** 2 fair
**Presentation:** 2 fair
**Contribution:** 2 fair
**Rating:** 3
**Confidence:** 4

**Summary:**

The authors propose a new architecture for emergent communication in environments without prior knowledge. The proposed method operates under a variation of the traditional Lewis Games where random noise is often added to the messages.

**Strengths:**

This paper analyses a variation of the popular Lewis Games, MRILG, where the agents can take advantage of more information coming from the others before taking an action. Additionally, in these games noise is added to the messages, which is important because communication is often noisy and it should be investigated how communication can be done under noisy conditions.

**Weaknesses:**

* This paper does not explore the type of messages learned by the agents. It would be interesting to visualize in some way some of the messages learned by the agents to communicate.
* The entropy term in the loss is not defined in the paper and the motivations for the use of this term do not seem good enough.
* In table 2, I wonder about the significance of these results; of course the agents are trained with noise, they will perform better in noisy testing than agents that were not trained with noise.
* The use of noise is framed as a major contribution. However, using random noise in the messages is not new; the authors give the example of the work with the zipfs law to analyse language properties (page 2) but this is not the only place where noise is used and analysed [1, 2, 4].

Minor:
* In page 3: "When there is no ambiguity, we drop the dependence of for $m(x; \theta)$": "of for" repeated
* In page 5: "and effective game round, $i \in {1, . . . , I}$" it seems that the number of rounds in now defined as $I$ while before in section 2.2 it was defined as $N$
* Throughout section 2, $x$ is referred to both as the original message sent by the speaker and as the prediction made by the listener
* There are too many complex equations written within the text. Some of them should be written instead in equation blocks as the way it is done becomes difficult to read.

Please find below related questions.

[1] https://arxiv.org/pdf/2010.15896.pdf

[2] https://openreview.net/pdf?id=O5arhQvBdH

**Questions:**

1. In section 2.2, it is unclear to me what is the $unk$ token and how the noise is applied. From equation (1) I understand it results from the noise function if $p\leq\lambda$, but I fail to understand where $unk$ comes from. Is it a fixed token? If so, I cannot agree that noise is being applied to the message, also because, according to the equation, the message $m$ is not affecting any of the generated noise. Could the authors elaborate on this?
2. It is unclear to me what $\hat{x}$ means. In page 3, it is both stated "where the goal is to try to identify the image $x â\in C$ that the Speaker received, $\hat{x}= x$" and "round when the Listener plays the I don't know (idk) action, \hat{x} = \hat{x}_{idk}". It seems to have different meaning in each case. In the first case it seems to denote the guess of the listener and in the second it seems to define an action. Could the authors clarify?
3. In page 6: "where we linearly increase the noise level in the communication channel from 0 to $\lambda$.". From equation (1) $\lambda$ represents a probability of wether insert noise or not. How does increasing $\lambda$ will increase the level of noise? While it can happen, it does not seem necessarily true that it will happen.
4. In tables 1 and 2, since LG(RL) and NLG are the same but NLG uses noise (as described in section 3.1); how is the accuracy of NLG much higher? is the existence of noise ($\lambda\geq 0$) beneficial for learning?
5. Is the variation LG(RL) a contribution (as stated in section 3.1)? REINFORCE as been used before in emergent communication games [3, 4]
6. do the authors allow the gradients to flow across agents as it happens in works such as DIAL [5], or are they fully independent?

Overall, I have several concerns regarding the contribution of this work and the approaches presented that I would like the authors to comment on.

[3] https://arxiv.org/pdf/1705.11192.pdf

[4] https://arxiv.org/pdf/1804.03980.pdf

[5] https://arxiv.org/pdf/1605.06676.pdf

---

> ### Author Response · Authors · 2023-11-23
> **Response to Reviewer zpnz**
>
> ## Weaknesses
>
> > **This paper does not explore the type of messages learned by the agents. It would be interesting to visualize in some way some of the messages learned by the agents to communicate.**
>
> We thank Review zpnz for the interesting suggestion. Since the agents learn an abstract language using abstract tokens, we do not have a good way to visualize the messages sent, but it could be an interesting future work direction.
>
> ---
>
> > **The entropy term in the loss is not defined in the paper and the motivations for the use of this term do not seem good enough.**
>
> The loss functions used in both agents are fully detailed in Appendix E.1. Additionally. The entropy term was fully evaluated in [Chaabouni et al., 2022]. In the text, we point to this paper for further details (see Section 2.2.2).
>
> ---
>
> > **In table 2, I wonder about the significance of these results; of course the agents are trained with noise, they will perform better in noisy testing than agents that were not trained with noise.**
>
> We can see a clear difference in the emergent communication protocol learned. Looking at Table 1 and Table 2, we can understand that the agents trained in the NLG (with noise) can create a robust communication protocol to handle noisy and noiseless messages.
>
> ---
>
> > ** The use of noise is framed as a major contribution. However, using random noise in the messages is not new; the authors give the example of the work with the zipfs law to analyse language properties (page 2) but this is not the only place where noise is used and analysed [1, 2, 4].**
>
> We thank Reviewer zpnz for the suggestions. We added these works to the extended related work section in Appendix A. Nevertheless, we maintain our claim for novelty for the following reasons. All works [1, 2, 4] simplify the problem by having a DIAL method, meaning gradient flows between agents. In our case, we implement a RIAL architecture where each agent sees the other as part of the environment, substantially increasing the problem's complexity. As such, the Speaker does not even know that the message is suffering modifications, making the coordination problem extremely complicated. [1] also uses a continuous communication problem, making it distant from the scope of our work.
>
> ---
>
> > **In page 5: "and effective game round, $i\in\{1,\ldots,I\}$" it seems that the number of $I$ rounds in now defined as while before in section 2.2 it was defined as $N$**
>
> There is a difference between $I$ and $N$. $I$ is the number of rounds played in a game and $N$ is the maximum number of rounds a game can have. We will make this distinction more clear, see Section 2.2.2.
>
> ## Questions
>
> > **In section 2.2, it is unclear to me what is the *unk* token and how the noise is applied. From equation (1) I understand it results from the noise function if, but I fail to understand where comes from. Is it a fixed token? If so, I cannot agree that noise is being applied to the message, also because, according to the equation, the message is not affecting any of the generated noise. Could the authors elaborate on this?**
>
> Yes, *unk* is a fixed token. We apply noise to the message by substituting a variable number of message tokens with the *unk* token. The noise applied is externally introduced in the message. The Speaker does not even know if the message contains noise, adding additional complexity.
>
> ---
>
> > **It is unclear to me what $\hat{x}$ means. In page 3, it is both stated "where the goal is to try to identify the image $\hat{x} \in \mathcal{C}$ that the Speaker received, $\hat{x}=x$" and "round when the Listener plays the I don't know (idk) action, $\hat{x} = \hat{x}_{idk}$". It seems to have different meaning in each case. In the first case it seems to denote the guess of the listener and in the second it seems to define an action. Could the authors clarify?**
>
> $\hat{x}$ is the action selected by the Listener. This action can be an image or the action ``I don't know'' ($x_{\text{idk}}$). If an image is selected by Listener, $\hat{x}$ is also one of the images of a set (candidates set) given to the Listener.
>
> > **In page 6: "where we linearly increase the noise level in the communication channel from 0 to $\lambda$.". From equation (1) $\lambda$ represents a probability of wether insert noise or not. How does increasing $\lambda$ will increase the level of noise? While it can happen, it does not seem necessarily true that it will happen.**
>
> For each token in the message, we draw a number from the uniform distribution $l\sim\mathcal{U}(0,1)$. If $l>\lambda$ we change the message token by the \textit{unk} token. In expectation the number of removed token is $\lambda\cdot T$, where $T$ is the message length.

---

> > ### Author Response · Authors · 2023-11-23
> > **Response to Reviewer zpnz (part 2)**
> >
> > > ** In tables 1 and 2, since LG(RL) and NLG are the same but NLG uses noise (as described in section 3.1); how is the accuracy of NLG much higher? is the existence of noise ($\lambda \geq 0$) beneficial for learning?**
> >
> > Yes, from table 1 and 2 we can see that when adding a noisy communication channel allows for the agent to process more diverse information and create robust communication channel that work in both cases (Lewis Game with and without noise).
> >
> > ---
> >
> > > **Is the variation LG(RL) a contribution (as stated in section 3.1)? REINFORCE as been used before in emergent communication games [3, 4]**
> >
> > We thank Reviewer zpnz for the suggestions. We added these works to the extended related work section in Appendix A. The architectures applied for the games we propose are novel. The proposed games use natural images as input and discrete messages. For example, in [3], the input has very few degrees of freedom; [4] uses a DIAL method, meaning gradient flows from the Listener to the Speaker, which implies severe architectural differences. In our case, we propose a new Listener architecture that uses an attention mechanism to combine information from the noisy message and each candidate's information. Additionally, we also modify the Listener's policy to take into account time-dependent environments, where we give the possibility to the Listener reason if it prefers to play another round (gather more information) or make a final decision.
> >
> > ---
> >
> > > **do the authors allow the gradients to flow across agents as it happens in works such as DIAL [5], or are they fully independent?**
> >
> > As described in the beginning of section 2 (and Appendix A), we impose a RIAL setting, no gradients flow between the Speaker and Listener.

---

### Official Review · Reviewer_7Viy · 2023-10-31

**Soundness:** 2 fair
**Presentation:** 3 good
**Contribution:** 2 fair
**Rating:** 3
**Confidence:** 4

**Summary:**

The paper introduces a novel multi-agent architecture for emergent robust communication, emphasizing a shared language in noisy environments. The research also presents a new referential game, enhancing complexity and interaction."

**Strengths:**

1. The research adeptly merges insights from human language evolution with artificial language development, leveraging deep learning to highlight the capabilities of neural agents in autonomous communication.
2. By innovatively adapting the Lewis Game with noise and time elements, the authors elevate its realism, creating a more comprehensive and challenging framework.
3. The paper's emphasis on developing robust communication protocols for noisy environments is both timely and crucial, addressing a pivotal challenge in the field.

**Weaknesses:**

1. The authors' reliance on established paradigms like the Lewis Game, even with modifications, raises concerns about true innovation. Is this just a repackaging of old concepts?
2. The paper lacks a rigorous empirical validation of its proposed architecture, leaving readers questioning its real-world applicability.
3. While the literature review is extensive, the paper falls short in critically analyzing the limitations of referenced works, resulting in potential oversights in the proposed methodology.
4. The emphasis on noise adaptation, though relevant, is hardly novel in the field of emergent communication. The authors fail to differentiate their approach sufficiently from existing solutions.
5. The comparative approach with Ueda & Washio 2021 feels superficial, lacking in-depth analysis on fundamental differences

**Questions:**

1. Given the modifications you've introduced to the Lewis Game, how do you justify that these changes bring genuine innovation in the context of multi-agent reinforcement learning, as opposed to simply adding complexity to an existing framework?
2. The concept of noise adaptation in emergent communication is not new. How does your approach fundamentally differ from existing solutions, and what unique challenges does it address in the multi-agent reinforcement learning landscape?
3. Your comparison with the Ueda & Washio 2021 study appears to lack depth. Could you elucidate the fundamental differences in the underlying assumptions, problem formulations, and outcomes between their work and yours, especially in the context of multi-agent dynamics?

**Details Of Ethics Concerns:**

No.

---

> ### Author Response · Authors · 2023-11-23
> **Response to Reviewer 7Viy**
>
> ## Weaknesses
>
> > **The authors' reliance on established paradigms like the Lewis Game, even with modifications, raises concerns about true innovation. Is this just a repackaging of old concepts?**
>
> The Lewis Game is a well-established task in emergent communication since it presents several crucial challenges to overcome to understand and study how communication can emerge to solve a cooperative task. Most related work presented in the paper and the related work proposed by all reviewers has experiments based on a derivation of the original Lewis Game. In our proposed work, we follow the work proposed by [Chaabouni et al., 2022] that complexifies the original Lewis Game by using huge natural image datasets for the input distribution and increases the number of candidates given to the Listener, increasing the difficulty of the discrimination task. The main focus of our work is to study and explore how noisy conditions can affect the emergence and quality of the communication protocol in such tasks. As described in Section 3, previous methods are not robust to noise in the communication channel. As such, we proposed new agent architectures and training schemes, creating robust communication channels.
>
> ## Questions
>
> > **Given the modifications you've introduced to the Lewis Game, how do you justify that these changes bring genuine innovation in the context of multi-agent reinforcement learning, as opposed to simply adding complexity to an existing framework?**
>
> We would like to emphasize that the primary purpose of our work is not to create a MARL system. The main trends in MARL focus on studying centralized training with decentralized execution, how partial observability affects performance, and how to exchange information through communication. In contrast, emergent communication studies aim to assemble cooperative tasks to study emergent communication and its properties.
>
> We now clarify the novelty of our work. First, we present a series of experimental evaluations that give new insights into how noise affects the emergence of communication when using more realistic datasets (in our case, ImageNet and CelebA). Secondly, we merge noisy and time-dependent environments, creating highly complex environments that generalize the original Lewis Game. Finally, we derive new agent architectures to play these realistic and more challenging games. For example, most (if not all) methods proposed in previous works could not take advantage of the Multi-Round Indecisive Lewis Game (MRILG) because these methods are not prepared to have time dependencies in their architecture.
>
> ---
>
> > **The concept of noise adaptation in emergent communication is not new. How does your approach fundamentally differ from existing solutions, and what unique challenges does it address in the multi-agent reinforcement learning landscape?**
>
> In the introduction, we talk about [Ueda and Washio, 2021]. Even though previous works consider noisy environments in emergent communication, the difference to our work is two-fold. First, we only consider discrete messages as [Ueda and Washio, 2021]. Second, much like  [Chaabouni et al., 2022], we take into consideration more realistic datasets (discriminate natural images) applied to the context of having noisy communication channels. We added to appendix a section for related work, see Appendix A.

---

> > ### Author Response · Authors · 2023-11-23
> > **Response to Reviewer 7Viy (part 2)**
> >
> > > **Your comparison with the [Ueda and Washio, 2021] study appears to lack depth. Could you elucidate the fundamental differences in the underlying assumptions, problem formulations, and outcomes between their work and yours, especially in the context of multi-agent dynamics?**
> >
> > Please check Appendix A, for a more in-depth comparison (extended related work).
> >
> > As explained in the introduction, the main difference is that [Ueda and Washio, 2021] promotes an adversarial setting. The noise is introduced by substituting a token with another one. In this setting, the main focus is to trick the Listener by giving another plausible message different from the one created by the Speaker. By creating a problem like this, the authors want to force a pair of agents to create messages as short as possible to test the ZLA hypothesis. Additionally, [Ueda and Washio, 2021] also simplifies the environment by giving as input one-hot encoded inputs to the Speaker.
> >
> > In our case, not only do we use a more realistic dataset increasing complexity, but we also focus on a different problem. Our focus is to create robust communication protocols that can handle external interference in the communication channel, for example, simulating the loss of information in the communication channel. As such, we apply noise by substituting tokens into a fixed and extra token called the unknown token. Another particularity of our work is that we want our method to function in the noiseless and noisy cases at the same time. We added an extended related work section to the appendix where we fully detail the differences between our work and [Ueda and Washio, 2021].
> >
> > Summarizing, in the underlying assumptions, we differ from [Ueda and Washio, 2021] since we assume that the noisy communication channel has external interference, which translates into the loss of information. On the other hand, [Ueda and Washio, 2021] assumes an adversarial setting where the objective is trying to deceive the Listener. Regarding the problem formulation, the game proposed by [Ueda and Washio, 2021] is a simpler version of our NLG game. In our case, we use natural images as input. Additionally, the Listener's discrimination task takes into account up to 4096 images to discretize, where in [Ueda and Washio, 2021] the input is a simple one-hot vector with a maximum size of 256. Moreover, since the input is one-hot encoded, the Listener does not need to receive candidates and just outputs a number between 1 and 256. For the expected outcomes, we show that our methods work on the noiseless and noisy case, creating robust communication protocols. On the other hand, [Ueda and Washio, 2021] only wants to force the communication protocol to create small messages by introducing adversarial noise, where a test between settings (train without noise and test with noise, and vice-versa) was never made or even intended.

---

### Official Review · Reviewer_WCrY · 2023-11-01

**Soundness:** 2 fair
**Presentation:** 2 fair
**Contribution:** 2 fair
**Rating:** 5
**Confidence:** 4

**Summary:**

The paper focus on creating a common language among agents to enable cooperation and solve tasks in noisy environments. Then the authors present a novel multi-agent architecture for learning a discrete communication protocol without prior knowledge of the task to solve. The authors introduce a referential game based on the Lewis Game, with added complexity of random noise in message transmission and multiple interactions between agents before making a final prediction. The proposed architecture demonstrates equivalent generalization aptitude to simpler games, while being the only method capable of producing robust communication protocols that handle cases with and without noise.

**Strengths:**

- The paper introduces a novel multi-agent architecture for learning a communication protocol without prior knowledge of the task, and it explores the challenges of noisy environments and multiple interactions, which adds originality to the field.
- The paper provides a detailed analysis of the learning strategy for both agents, and it presents a comprehensive architecture with different modules for processing messages and images.
- The paper's contributions are significant as it addresses the goal of creating a common language among agents for cooperation and solving tasks. It also explores the impact of noise and demonstrates the ability to produce robust communication protocols. The findings have implications for understanding emergent communication and its applications in challenging environments.

**Weaknesses:**

However,

- The paper could benefit from comparing the proposed architecture to existing approaches in the field of emergent communication. This would provide a better understanding of the novelty and effectiveness of the proposed method.
- The evaluation of the proposed architecture is focused on the newly developed referential game. It would be valuable to evaluate the architecture on a wider range of tasks and compare its performance to other architectures to assess its generalizability.
- While the paper mentions the capability of the proposed architecture to handle noise, there is limited discussion on how the architecture specifically addresses and mitigates the impact of noise in the communication channel. Providing more details on this aspect would enhance the clarity and understanding of the proposed method.
- Ablation studies, where different components or modules of the architecture are systematically removed or modified, could provide insights into the contribution and importance of each component. This would strengthen the analysis and understanding of the proposed architecture.

**Questions:**

Please see Weaknesses

**Details Of Ethics Concerns:**

No ethics concerns.

---

> ### Author Response · Authors · 2023-11-23
> **Response to Reviewer WCrY**
>
> ## Weaknesses
>
> > **The paper could benefit from comparing the proposed architecture to existing approaches in the field of emergent communication. This would provide a better understanding of the novelty and effectiveness of the proposed method.**
>
> The main focus of our work is to introduce a new approach to handle external interference in the communication channel. We present a comparative analysis with [Chaabouni et al., 2022], which proposes the original LG (SS) (see section 3), where the Listener implementation is a self-supervised agent. We demonstrate in section 3 that our proposed architecture, where the Listener is an RL model, is crucial to developing robust communication protocols that can handle the noiseless and noisy case.
>
> Additionally, we argue that our work is different enough from previous approaches where a direct comparison could be meaningless or time-consuming, where severe alterations are needed to modify previous architectures. We now elaborate on our argument. We can divide the emergent communication literature into two major clusters. On one set, the works employ discrete communication channels, forcing a RIAL implementation where each agent learns its parameter set and treats other agents as part of the environment. The architectures proposed in [Lazaridou et al., 2016,
> Choi et al., 2018, Bouchacourt and Baroni, 2018, Lazaridou et al., 2018, Graesser et al., 2019,
> Li and Bowling, 2019, Ueda and Washio, 2021] do not consider time dependencies. As such, to compare such architectures, we need to make substantial architectural changes to make sure agents can go through multiple rounds.
>
> On the other hand, we have works that simplify the problem of emergent communication where the gradient can flow between agents (DIAL methods) [Havrylov and Titov, 2017, Mordatch and Abbeel, 2018, Tieleman et al., 2019,
> Guo et al., 2019, Chaabouni et al., 2020, Rita et al., 2022]. This oversimplification of the problem is unnatural when looking from the viewpoint of simulating human communication, which is non-differentiable. From this standpoint, comparing our approach with such works appears inappropriate because the introduction of noise is not straightforward. In the naive case, the noise would have almost no negative impact since gradients flow through agents, where protocols are robust by default. For example, a VAE architecture is robust to noise since the gradient flows freely from the decoder to the encoder, and the latent space is continuous.
>
> ---
>
> > **The evaluation of the proposed architecture is focused on the newly developed referential game. It would be valuable to evaluate the architecture on a wider range of tasks and compare its performance to other architectures to assess its generalizability.**
>
> In Appendix F, we introduce and evaluate the agents trained in every Lewis Game variation in other transfer learning tasks with the main purpose of testing their capability to generalize to new tasks.
>
> Additionally, the vast majority of works in emergent communication are based fundamentally only on this task - a variation of the Lewis Game. The LG is a simple but highly customizable task that evaluates several properties of emergent languages, allowing the connection to human languages.
>
> ---
>
> > **While the paper mentions the capability of the proposed architecture to handle noise, there is limited discussion on how the architecture specifically addresses and mitigates the impact of noise in the communication channel. Providing more details on this aspect would enhance the clarity and understanding of the proposed method.**
>
> In section 3.2, we motivate that our novel agent architecture perform better than the baseline. In section 3.2.1, we demonstrate that if we generalize the Lewis Game, where the communication channel can modify messages by introducing noise, our proposed architecture can still be robust and have high performance in both games (Lewis Game without noise and Lewis Game with noise). The second architectural change appears in section 2.2, where we define the MRILG (multi-round indecisive Lewis Game) and further extend the Listener architecture to contain an extra action, the *I don't know* action. When the Listener chooses this action, the pair of agents play a subsequent round in the Lewis Game, which allows the Listener to receive and process more information.
>
> To give some additional information, the original Lewis Game (LG) is played only in one round - the Speaker receives an image and creates a message describing the image. The Listener tries to guess the image received by the Speaker over a set of candidate images taking only the message into account. In MRILG, several rounds of the LG can be played as long as the Listener chooses the idk action. During these several rounds, the Speaker's input image and the candidates' set are always the same.

---

> > ### Author Response · Authors · 2023-11-23
> > **Response to Reviewer WCrY (part 2)**
> >
> > > **Ablation studies, where different components or modules of the architecture are systematically removed or modified, could provide insights into the contribution and importance of each component. This would strengthen the analysis and understanding of the proposed architecture.**
> >
> > Due to space constraints, ablation studies were left in the Appendix. Precisely, we focus on the Listener's attention mechanism, where we compare how introducing non-linearity affects the overall performance (Appendix D). Additionally, in Appendix E.4, we ablate different head configurations, taking into account the actor and critic heads.

---

### Meta-Review · Area_Chair_v4m7 · 2023-12-14

**Metareview:**

This paper designs new multi-agent architecture capable of learning a discrete communication protocol without any prior knowledge of the task to solve. This paper also creates a new referential game based on the original Lewis Game to make tasks more difficult. The paper shows empirically several advantage of the proposed architecture when evaluating on the newly developed games. While the paper introduces a few new concepts/architecture, reviewers remain concerned with the novelty of newly proposed game (similar types of modification appeared earlier), and the several important details of experiments especially regarding to how the architecture handle noise and whether designed experiment is enough to show that “robust communication” emerge. Therefore, we recommend rejection.

**Justification For Why Not Higher Score:**

novelty/contribution is limited. Several details of experiment may need to be improved.

**Justification For Why Not Lower Score:**

N/A

---

### Decision · Program_Chairs · 2024-01-16

Reject